# Molecular Dynamics Studies of Hydrogen Effect on Intergranular Fracture in *α*-Iron

**DOI:** 10.3390/ma13214949

**Published:** 2020-11-04

**Authors:** Xiao Xing, Gonglin Deng, Hao Zhang, Gan Cui, Jianguo Liu, Zili Li, Bingying Wang, Shouqin Li, Chao Qi

**Affiliations:** 1College of Pipeline and Civil Engineering, China University of Petroleum (East China), Qingdao 266500, China; d1441418858@163.com (G.D.); chennacuigan@163.com (G.C.); jgliu83@163.com (J.L.); 2Department of Chemical and Materials Engineering, University of Alberta, Edmonton, AB T6G 1H9, Canada; hao7@ualberta.ca; 3School of Materials Science and Engineering, China University of Petroleum (East China), Qingdao 266580, China; tdwby2004@126.com; 4Dongxin Oil Production Plant, Shengli Oilfield Company SINOPEC, Dongying 257094, China; lishouqin861.slyt@sinopec.com; 5Storage and Transportation Branch of China North Huajin Chemical Industry Service Co., Ltd., Panjin 124010, China; lian59455@163.com

**Keywords:** hydrogen embrittlement, intergranular cracking, atomistic mechanism, molecular dynamics

## Abstract

In the current study, the effect of hydrogen atoms on the intergranular failure of *α*-iron is examined by a molecular dynamics (MD) simulation. The effect of hydrogen embrittlement on the grain boundary (GB) is investigated by diffusing hydrogen atoms into the grain boundaries using a bicrystal body-centered cubic (BCC) model and then deforming the model with a uniaxial tension. The Debye Waller factors are applied to illustrate the volume change of GBs, and the simulation results suggest that the trapped hydrogen atoms in GBs can therefore increase the excess volume of GBs, thus enhancing intergranular failure. When a constant displacement loading is applied to the bicrystal model, the increased strain energy can barely be released via dislocation emission when H is present. The hydrogen pinning effect occurs in the current dislocation slip system, <111>{112}. The hydrogen atoms facilitate cracking via a decrease of the free surface energy and enhance the phase transition via an increase in the local pressure. Hence, the failure mechanism is prone to intergranular failure so as to release excessive pressure and energy near GBs. This study provides a mechanistic framework of intergranular failure, and a theoretical model is then developed to predict the intergranular cracking rate.

## 1. Introduction

Despite the rapid development of new materials, steel is still irreplaceable in the energy industry, especially in oil and gas transmission [1]. However, steel is susceptible to rupture under the influence of hydrogen atoms generated from cathodic reactions or the service environment [2]. Hydrogen embrittlement (HE) causes the fracture mode of materials to change from ductile dimple fractures to brittle transgranular or intergranular cracking [3]. This seriously affects the usability of materials and causes great damage in the engineering application of iron and steel materials. Therefore, over many decades, scholars all over the world have been continuously conducting research on hydrogen embrittlement. Many mechanisms, such as the hydrogen-enhanced decohesion theory (HEDE) [4,5,6], hydrogen-enhanced local plasticity (HELP) [7,8,9,10], and the hydrogen bubble theory [11,12,13], have been proposed to explain the phenomenon of hydrogen embrittlement, and a large number of databases on the failure of metal structures caused by hydrogen are available. Although considerable efforts have been made to study the atomistic mechanism of hydrogen-assisted cracking, there is no universal conclusion regarding the intergranular and transgranular cracking mechanism. Moreover, no statistical model is available to estimate intergranular crack growth when the treated metal is used in a hydrogen environment.

As a basic understanding, the cracking mode of steel is predominantly transgranular. The quasi-cleavage phenomenon was observed by Birenis et al. [14], who reported that hydrogen atoms restricted dislocation slipping bands and proved that the hydrogen disbanding mechanism might be the exact explanation for hydrogen embrittlement. Hydrogen atoms are considered to weaken the cohesive energy of the grain boundary (GB), thus inducing intergranular failure. Bechtle et al. [15] verified that the susceptibility of HE in steel could be tremendously increased with an increase in the free volume in GBs. Extensive studies have also indicated that an excessive volume of GBs can trap hydrogen atoms and hinder hydrogen transportation [16]. Correspondingly, Teus et al. show that the diffusion coefficients of H along the GB are much smaller than that of bulk diffusion, and this verifies that hydrogen atoms are prone to be trapped in an excess volume of GBs [17]. Sibata et al. examined the crystallographic orientation beneath the fracture surface and showed that microcracks close to the crack surface were parallel to the {110} planes [18,19], which inferred that crack propagation was prone to the specific orientation of the crystal and exhibited intergranular features. Those results implied that if hydrogen atoms’ expansion effect on GBs was verified, the intergranular failure might be facilitated by hydrogen segregation. To understand GB engineering, quantitative models that considered hydrogen movement were developed in order to predict the probability of intergranular failure [13,20]. In those studies, mobile hydrogen atoms would continuously diffuse to the GBs and increase the probability of intergranular failure. Each of these proposed mechanisms and models is supported by many experimental and theoretical works, but the multifaceted nature of hydrogen-plastic deformation interactions complicates the application of any of those mechanisms. Furthermore, experimental in-situ observations of GB failure are not available with current technology, and the nature of hydrogen effects on intergranular cracking remains unclear. 

To study the GB failure mechanism, molecular dynamics (MD) were applied in the current study. A series of low ∑tilt GBs in which the ∑value corresponded to the reciprocal ratio of the coincidence site lattice were examined. The correlations of the excess volume and hydrogen insertion-induced strain energy of GBs were quantified, and a theoretical model was established to predetermine the cracking probability in GBs.

## 2. Simulation Methodology

The bicrystal model that is investigated in this study is shown in Figure 1a; the model consists of two grains rotated along the [110] tilt axis, which is parallel to the z-axis. The symmetric tilt grain boundaries are generated by tilting the upper and lower grains to the same angles but in opposite directions, as shown in Figure 1b. Simulation Cells that were comprised of 54,000 to 58,000 Fe atoms were (97~107) × (193~227) × 28 Å^3^ in three dimensions. Because a periodic boundary condition is applied in all three directions, there are two identical GBs in each bicrystal model. An isothermal-isobaric (NPT) ensemble [21,22] with velocity rescaling was first applied to maintain the system’s pressure and temperature in order to construct the GBs’ equilibrium structure. In particular, the bicrystals were first constructed at room temperature, then heated to 700 K and maintained at 700 K for 250 ps to reach equilibrium, and finally cooled down to 300 K. After the relaxation of the GBs, uniaxial displacement deformation was applied by fixing rigid layers within 10 Å of the top and bottom in the y-direction with a displacement rate of 0.024 Å/ps in all bicrystal systems. Zero pressure was applied in the x- and z-directions [23,24] during deformation, and a constant temperature was maintained using the Nose–Hoover method [22,25]. Atomic interactions were described using the embedded atom method (EAM) [26,27]. The Fe potential was based on the Mendelev form of EAM, the Fe-H interaction being fitted with density functional theory (DFT) calculations. The fitted Fe-H potential had been applied to calculate the free surface energy, stacking fault energy, and strain-stress curves. The simulation results matched well with the experimental tests. Moreover, the fitted potential had been verified by several previous works [27,28]. All the simulations were carried out with LAMMPS (Development version), and the OVITO (3.3.0) was applied for visualization.

## 3. Results Discussion

### 3.1. Hydrogen Effects on GB Energy and Excess Volume 

The bicrystal models were first heated from 300 K to 700 K. The heating rate remained constant at 8 K/ps, and the model was maintained at a maximum temperature for 250 ps. The cooling rate was the same as the heating rate, and when the temperature returned to 300 K, the system was relaxed for another 250 ps to ensure that the system reached equilibrium. The GB energy (GBE) was then calculated using the following equation [28]:(1)$$\gamma_{GB}=\frac{E_{bicrystal}-ne_{Fe}}{2A_{GB}}$$
where *E**_bicrystal_* is the total energy of atoms in the bicrystal system at 300 K, *n* is the total number of iron atoms in the system, *e**_Fe_* is the cohesive energy of Fe in the BCC structure at 300 K, and *A**_GB_* is the grain boundary area. Because of the disordered structure of the GBs, the volume of the bicrystal model is larger than that of the single crystal model that consists of the same number of atoms. The volume expansion that is caused by the GB is denoted as Δ*l* and can be calculated as: (2)$$\Delta l=\frac{V_{bicrystal}-nv_{Fe}}{2A_{GB}}$$
where *V_bicrystal_* is the total volume of the bicrystal system, *n* is the total number of Fe atoms in the bicrystal system, and *v**_Fe_* is the atomic volume of a single iron atom in the BCC structure at 300 K. The relationship between volume expansion and GBE is shown in Figure 2. 

The comparison suggests that grain boundary expansion is not necessarily positively related to GBE, and no linear correlation has been observed. The ∑3 GB possessed the lowest volume expansion and GBE value simultaneously, which suggests that it was the most stable GB when no hydrogen was present. In order to study the effect of hydrogen on GBE and GB expansion, hydrogen atoms were then introduced into the system randomly at an atomic concentration 0.001 to investigate the hydrogen segregation and hydrogen potential in the GBs. The systems were maintained at 300 K with no load to relax for 4 ns. The hydrogen distribution in two typical GBs at times 0.1, 1, and 4 ns is shown in Figure 3. The hydrogen atoms in the GB possessed a lower potential energy at 0.1 ns, which indicated that they were comparatively stable when compared with those in bulk, and there were trapping site series with different energies in all GBs. In the low volume ∑3 GB, the hydrogen energy was correspondingly larger. Comparatively, the hydrogen energy in the large volume ∑57 GB was smaller compared with that in the ∑3 GB. The energy difference indicates that the hydrogen atoms favorably exist in the GB with a larger volume. In all cases, hydrogen atoms continuously diffused to the GB and finally distributed within ±5 Å from the GB plane at 4 ns. In the following diagrams, the average hydrogen atoms’ potential energy is related to the GB where they are located. Next, we will examine the diffusivity and potential energy of hydrogen atoms versus the GB’s excess volume.

The total mean square displacement (MSD) over *N* hydrogen atoms was calculated according to the equation [16], in which *d* represents the dimension of the models, *N* is the total number of hydrogen atoms, *r*(*t_o_*) is the original location of hydrogen, and *r*(*t_o_* + Δ*t*) is the location of hydrogen at time *t_o_* + Δ*t*:(3)$$MSD=\frac{1}{2dN}\sum_{1}^{N}{\left|r\left(t_{o}+\Delta t\right)-r\left(t_{o}\right)\right|}^{2}$$

Figure 4 shows the MSD of hydrogen atoms in different GBs at 300 K. Diffusivities can be calculated by obtaining the slope of MSD curves versus time. Diffusivities of H atoms ranged from 2.7 × 10^−7^ cm^2^/s to 5 × 10^−7^ cm^2^/s in a rectangular region with a size of ±10 Å from the GB plane. Meanwhile, at room temperature, the average diffusivity value in pure and un-deformed α-iron is about 1 × 10^−4^ cm^2^/s [16]. This comparison verifies that hydrogen atoms tend to be trapped by grain boundaries. Moreover, the diffusivity of hydrogen atoms is bigger in GBs with a larger excess volume.

The correlation of the extra volume of GB and the average potential of H atoms in this GB is shown in Figure 5a. The ∑3 GB possesses the smallest excess volume, and the hydrogen atoms possess the largest potential energy. The ∑11 GB possesses a medium excess volume, and the hydrogen atoms possess a medium potential energy, while the ∑57 GB possesses the largest excess volume and the smallest potential energy. The extra volume of specific GB versus the diffusivity of H atoms in it is shown in Figure 5b. Even though the diffusivity of H atoms in GBs is comparatively small, the correlation verifies that H diffusivity in GBs is positively related to extra volume. These comparisons demonstrate that hydrogen atoms trapped in a smaller excess volume are unstable and have a higher potential energy. It could be inferred that such hydrogen atoms might have a more significant effect on GB deformation.

The previous study suggested that excess volume was closely related to the stability of a GB [29]. After hydrogen atoms were randomly inserted into the tetrahedral sites [30,31] within a rectangular zone (which was 10 Å away from the GBs), the system was relaxed for 2 ns to allow hydrogen atoms to segregate into the grain boundary region. As all the hydrogen atoms initially placed near GBs would segregate into the GBs, the concentration of hydrogen atoms in the same set of tests was consistent. The excess volumes of GBs were progressively measured at different atomic hydrogen concentrations (H/Fe), and the results are shown in Figure 6. The simulation results suggest that the presence of hydrogen atoms in the grain boundary can further increase the grain boundary excess volume, and this might weaken the bonding between iron atoms in the GB region and lead to intergranular cracking. At a small concentration of H, e.g., an atomic ratio of H/Fe of 0.005, the expansion is not significant; however, with an increase in the hydrogen concentration, the expansion effects become appreciable in all GBs. In particular, in ∑3, ∑57, and ∑33 GBs, the expansion ratio is positively linearly correlated with the hydrogen concentration, and this suggests that the GB expansion modulus might be constant.

The Debye Waller factors (DWFs) [32] of iron atoms, which are the measured square displacements (MSDs) at one ps, are used to verify the volume expansion after the hydrogen atoms are inserted into the GB. DWF values of all the atoms were tested after the models were relaxed for 2 ns at 300 K. DWF is an indirect measure of the atomic “cage size”, which is related to the free volume. If the cage size is large, the local structure is unstable, and cleavage fracture is more likely to occur. Hence, a larger DWF usually means a higher local free volume [33]. In Figure 7, the DWF values of the GBs are denoted in different colors; red indicates 0.07 Å^2^ or larger, and blue indicates 0.03 Å^2^ or less. As there is no hydrogen in the model, the ∑3 GB has a little extra volume, and the thickness is only two atomic layers. In contrast, the thickness of the other GBs is more extensive. When hydrogen atoms are inserted, the thickness and DWF average of all the GBs become bigger. The enlarging effect of hydrogen atoms on the ∑3 GB is the most significant as the hydrogen concentration is raised to 0.01, which verifies that hydrogen atoms have a more significant effect on GB deformation in a smaller free volume GB. Since the color of GB changes from mostly blue (0.02 Å^3^) to partially yellow or red (0.07 Å^3^) after hydrogen is inserted, the hydrogen enlargement effect has been verified in all GBs. The swelling effect will enlarge the distance between adjacent iron atoms, which infers that the cleavage fracture is more likely to occur between iron atoms. Even though the expansion effects vary in different GBs that may result from different failure mechanisms, the expansion effects are significant for all GBs as the hydrogen concentration rises from zero to 0.01.

Uniaxial displacement deformation was applied to the bicrystal model, and the GBs with a discrete excess volume exhibited different failure mechanics. A comparison of the strain-stress curves of the ∑3, ∑11, and ∑57 GBs is shown in Figure 8, where we obtained the engineering strain values and the average stress of iron atoms in rectangular regions that were 20 Å from the frozen loading layers. The yield strength of the ∑3 GB is larger, which verifies that it has a better stability. Furthermore, the modulus of the GBs ranged from 150 GPa to 210 GPa. Furthermore, the modulus of the GBs is negatively related to the excess volume. The ∑3 possesses the biggest modulus and the smallest excess volume, while the ∑57 possesses the lowest modulus and the largest excess volume. There is a considerable stress drop for each GB, which indicates the occurrence of large-scale plastic deformation or failure.

The hypothesis that the H enlargement of the volume in the GB results in intergranular failure has been verified in the ∑9 and ∑57 GBs, as shown in Figure 9. At a strain of 0.064, the microcrack initiates in the ∑9 GB with H, while the microcrack initiates at a strain of 0.066 in the no-H sample. At a strain of 0.09, the H sample’s crack length is much larger than that of the no-H sample. Additionally, no further crack propagation is recorded in the no-H sample, while a complete fracture occurs in the H sample at a strain of 0.091. A similar crack initiation and propagation process have been detected in ∑57 GB. When H exists, the intergranular crack initiates at a 0.063 strain, and a considerable crack propagation is recorded at a strain of 0.095. When no H exists, the void generates in the GB at a strain of 0.07, and no further crack generation can be detected. The results demonstrate that intergranular cracking occurs at a smaller strain because of hydrogen segregation in the GBs. Moreover, the microcrack in the GB is blunt when no hydrogen exists, while hydrogen activates the blunt microcrack in the GB and makes it brittle.

The hydrogen-facilitated phase transition and twinning mechanisms have been demonstrated in the other symmetric tilt GBs as shown in Figure 10. In the ∑3 and ∑27 GBs, a phase transition from body-centered cubic (BCC) to face-centered cubic (FCC) occurs under extreme stress [34,35]. For instance, in the ∑3 grain boundary, the BCC structure is indicated in green, the FCC structure is indicated in blue, and the unknown structure is indicated in red. When no hydrogen exists, the phase transition occurs at a strain of 0.079. After hydrogen atoms are introduced into the system, the phase transition starts to occur at a smaller strain of 0.074. In the ∑11 and ∑33 GBs, twinning is observed, which is related to the release of elastic energy when hydrogen exists. For example, in the ∑11 grain boundary, when no hydrogen exists, the dislocation emission from GB is the dominating energy dissipation mechanism for releasing the loading energy, and no twinning is observed from the GB during the loading. When hydrogen exists, the twinning is generated at a strain of 0.057. 

Apparently, in all GBs investigated here, the emission of first dislocations was inhibited due to the presence of hydrogen, and a comparison of dislocation emission strains with and without hydrogen is shown in Figure 11. Murakami et al. prove that hydrogen can play two roles in dislocation mobility: pinning (or dragging) and enhancement of mobility [36]. Our previous research also demonstrates that the hydrogen effect on dislocation nucleation occurs in accordance with its slip system: “the stacking fault energy of $\left[111\right]\left(\bar{1}\bar{1}2\right)$ slip system is increased with hydrogen concentration. In contrast, the stack fault energy of $\left[\bar{1}11\right]\left(110\right)$ is decreased with hydrogen concentration. Namely, the pinning effect happens in the $\left[111\right]\left(\bar{1}\bar{1}2\right)$ slip system, and the enhancement of mobility happens in the $\left[\bar{1}11\right]\left(110\right)$ slip system” [37,38]. Tong et al. clarify that the slipping system of the current bicrystal model is primarily <111>{112} [21]. Hence, the hydrogen pinning effect would occur in the current simulation system.

In the absence of H atoms, the loading energy is released via dislocation emission, which is followed by the phase transition, twinning emission, and cracking. Cracking is the least preferable and the most destructive way to release loading energy. Typically, the cracking mode should depend on the physical properties of the GB. The low energy ∑3 GB releases loading energy via plastic deformation emission, and no intergranular microcracks are formed, even when hydrogen atoms are present. Cracks generate in the GBs with a large excess volume, such as ∑57 GB. However, no further propagation is detected in the absence of hydrogen atoms. When hydrogen atoms are present, the dislocation emission from GB is restricted due to the fact that hydrogen atoms hinder the formation of dislocation via an increase in the stacking fault energy [39,40]. Consequently, the other three energy release mechanisms become more accessible when dislocation emission is suppressed. Notably, in ∑57 and ∑9 GBs, the microcracks form under smaller strains and can continuously propagate along the grain boundary. It was believed that hydrogen atoms enhanced cracking because of the hydrogen-induced free surface energy reduction. Since the volume expansion in different GBs is significant due to the segregation of hydrogen atoms (see Figure 6), it is expected that the deformation mode change, i.e., from ductile with no hydrogen to a cracking formation with hydrogen, can be quantitatively related to the volume expansion. 

### 3.2. Theoretical Model of Hydrogen Inducing Volume Expansion in GBs 

Based on the above simulation results, we conclude that if the hydrogen concentration is large enough, the strain energy is ultimately released via cracking in specific GB orientations. Thus, the increased strain energy is primarily released via the opening of free surfaces at the GB. The energy change caused by inserting H atoms into the iron lattice can be deduced from the H chemical potential [41] in iron, which can be expressed as follows: (4)$$u=u_{o}+k_{B}T\ln\left(\frac{c_{H}}{1-c_{H}}\right)+P\Omega$$

Here, *u* is the chemical potential of the H atom in iron, *u**_o_* is a constant, *k**_B_* is the Boltzmann constant, *T* is the temperature, *c**_H_* is the concentration of H (a constant for specific loading and environments), and *P*Ω is the interaction of the pressure field at the GB and the partial volume of the interstitial H atom in Fe. In a BCC iron study, the Ω value of H in the BCC iron structure was 1.9 × 10^−30^ m^3^ [42]. Thus, Δu=ΩδP, and the local pressure *P* is determined from the volume change, where $\delta P=k\frac{\delta V_{H}}{V_{o}}$ [43]. The change in strain energy that results from the segregation of hydrogen atoms into the GB can be expressed as follows:(5)$$\Delta u=k\Omega \frac{\delta V_{H}}{V_{o}}$$
where *V**_o_* is the original size of the bicrystal system, $\delta V_{H}$ is the volume expansion caused by hydrogen insertion, and *k* is the grain boundary bulk modulus, which are the slopes in the elastic region in Figure 8. For different GBs, the *k* value should not be the same. Here, for simplicity while not losing generality, we take the minimum and maximum *k* values of 150 and 210 GPa to calculate the GB energy changes that are caused by hydrogen segregation (Δu). Because the area of the model is fixed, the volume expansion can be denoted as δlHlo, where *l_o_* is the original length of the model. Therefore, the expansion ratio can be determined from Figure 6. Thus, the energy change can be quantified. If the entire change in strain energy that is caused by hydrogen segregation is the cost of creating voids in the GB, the area of the free surfaces of the void can be determined from the equation $\Delta u=2\gamma_{s}\Delta S$, where *γ**_s_* is the free surface energy and varies for the different crystallographic planes. The free surface energy for different orientations varies from 500 to 2000 mJ/m^2^, the average value of 1250 mJ/m^2^ is taken here to simplify the calculation [37,44], and Δ*S* is the free surface area, which is generated via hydrogen expansion and can be expressed as follows:(6)$$\Delta s=k\Omega \frac{\delta l_{H}}{2\gamma_{s}l_{o}}$$

The produced free surfaces are covered with hydrogen atoms, and each H atom occupies an area equal to (38π)23ao2, where *a_o_* is the lattice parameter of BCC iron. The total hydrogen number for generating the void can be expressed as N=(8π3)23ΔSao2. Hydrogen atoms diffuse into the GB from a spherical region around the GB, and the radius of the hydrogen supply region is determined by the local hydrogen concentration. The relationship for this is expressed as follows:(7)8ao3πr3cH3=(8π3)23ΔSao2
where *r* is the size of the hydrogen supply cracking zone around the void, and *c**_H_* is the atomic hydrogen concentration in the bulk of 0.0001 [45]. The velocity of the hydrogen atom at the GB depends on the diffusivity value, and the energy concentration can be expressed as $V=\frac{D\Delta u}{k_{B}T}$ [46], where *D* is the diffusivity of H atoms and has a value of 2 × 10^−9^ m^2^/s [47,48]. The time required to generate the free surfaces is *t* = *r*/*V*. The predicted cracking rate at the GB is equal to the size of the cracking zone over the hydrogen traveling time (2*r*/*t*), and this value ranges from 6.1 × 10^−10^ m/s to 8 × 10^−10^ m/s. The outcomes are consistent with experimental active crack rates of 3 × 10^−10^–1 × 10^−8^ m/s [2,49], and this demonstrates the brittle feature of intergranular crack growth. Although the predicted model and the mechanism proposed here do not explain every experimental outcome, the identified and quantified mechanism is consistent with the qualitative concepts that are advanced to explain the experimental results. 

## 4. Conclusions

The current study investigated the brittle feature of intergranular cracking. Typically, dislocation emission is the predominant way of releasing loading energy on GBs, and crack failure is not likely to occur at a comparatively small strain. However, hydrogen effects on the volume expansion of GBs and the inhibition of dislocation emission are demonstrated to induce intergranular cracking and to enhance phase transformations or twinning generation at the GBs. The hydrogen atoms could also initiate the dormant crack in the GBs.

This study clarifies fundamental physical rules that govern hydrogen interactions with GBs in iron that has a BCC structure. Hydrogen atoms trapped in a smaller excess volume are unstable and have a higher potential energy. It is also verified that such hydrogen atoms have a more significant effect on GB deformation. The hydrogen effect on GB dislocation emission is slip-system-based, the pinning effect happens in the $[111]\left(\bar{1}\bar{1}2\right)$ slip system, and the enhancement of mobility happens in the $\left[\bar{1}11\right]\left(110\right)$ slip system. The slipping system of the current bicrystal model is primarily <111>{112}, which is where the pinning effect occurs.

A comprehensive model is established to predict the intergranular cracking rate, given the proposal that the entire increase in the strain energy in the GB is required to create free surfaces that are covered by H atoms. The rate of free surface generation is thus related to the rate of hydrogen diffusion to the GB. This model provides a long-sought mechanistic insight that is crucial for understanding hydrogen-induced damage in structural materials.

## Figures and Tables

**Figure 1 materials-13-04949-f001:**
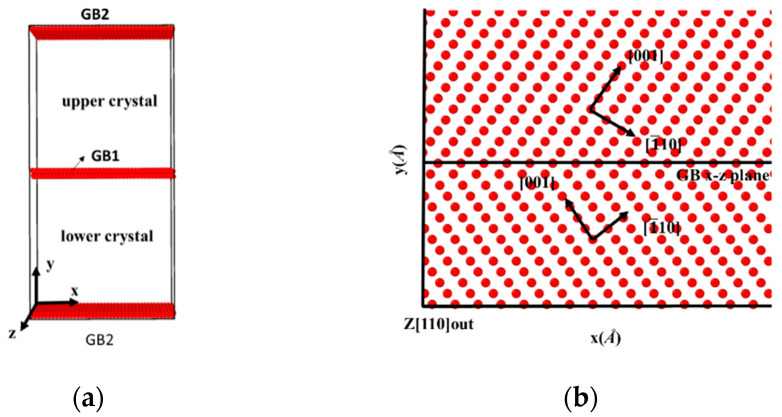
(**a**) Schematic of the bicrystal model, with two identical GBs in the model because the periodic boundary condition is applied in the y-direction. (**b**) Atomic configuration of a symmetrically tilted GB; the upper and lower crystals are rotated at the same angle but in opposite directions.

**Figure 2 materials-13-04949-f002:**
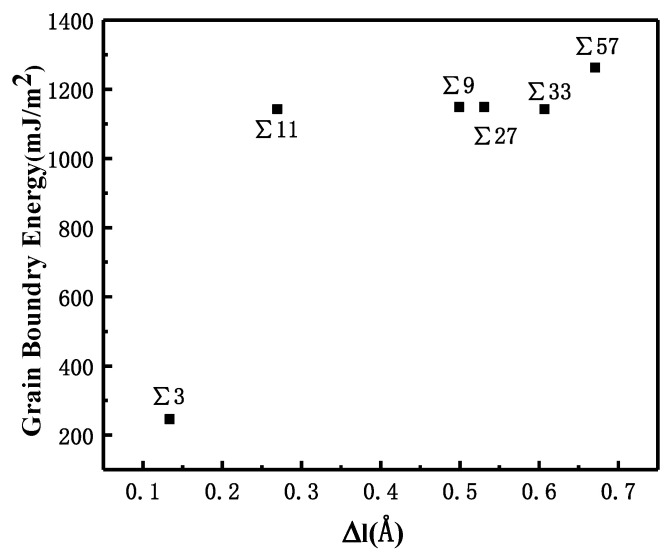
Correlation between GBE and volume expansion for different symmetric GBs.

**Figure 3 materials-13-04949-f003:**
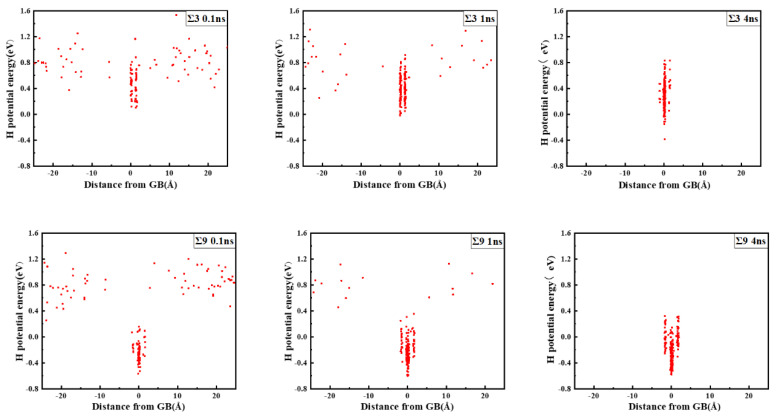
Hydrogen distribution and potential energy in the GBs. The zero value in the X axis indicates the center of the GB plane.

**Figure 4 materials-13-04949-f004:**
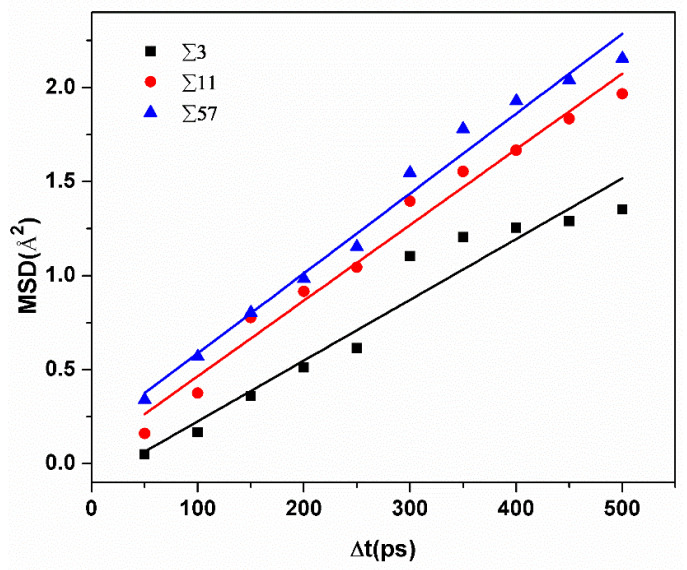
The comparison of the average MSD versus time of hydrogen atoms in different GBs.

**Figure 5 materials-13-04949-f005:**
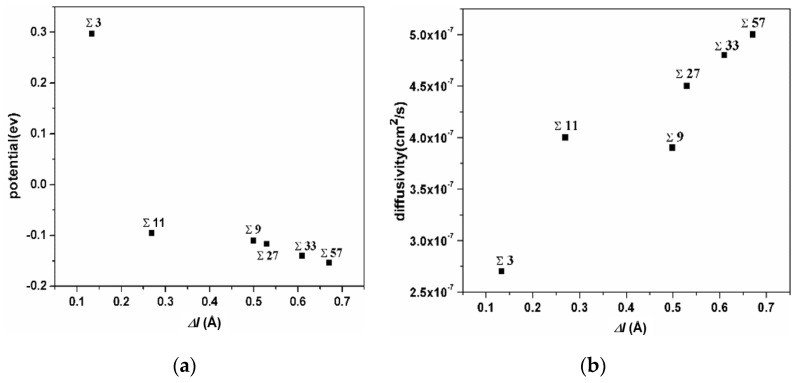
(**a**) the correlation of the extra volume of a specific GB versus the average of the potential energy of H atoms in it; (**b**) the correlation of the extra volume of a specific GB versus the diffusivity of H atoms in it.

**Figure 6 materials-13-04949-f006:**
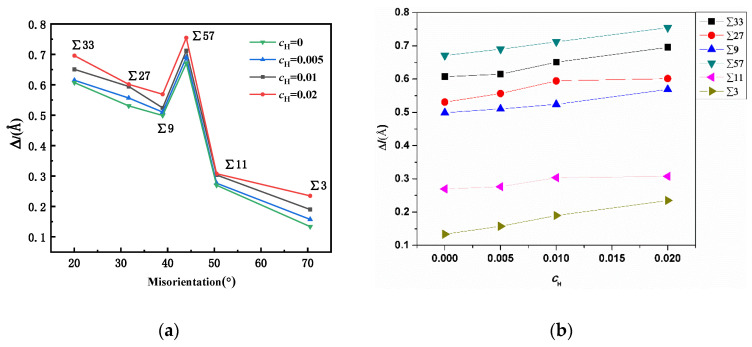
(**a**) Comparison of GB expansion at different hydrogen concentrations. (**b**) The hydrogen variation versus the excess volume change. The positive relation between the hydrogen concentration and the excess volume change is verified.

**Figure 7 materials-13-04949-f007:**
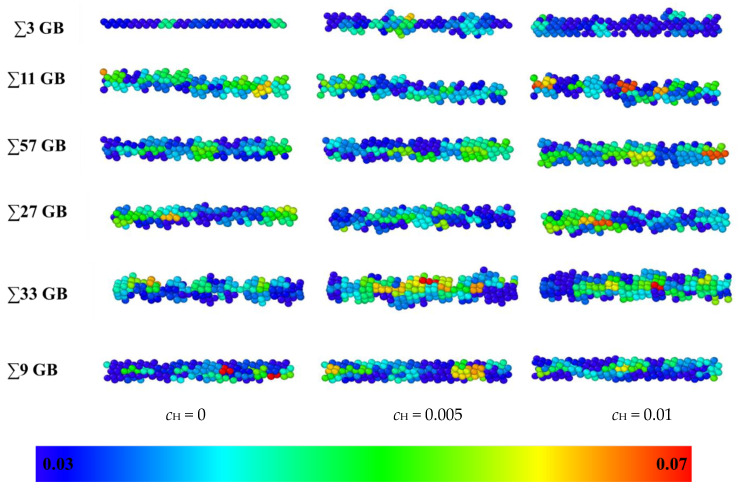
A comparison of the excess volume of different GBs at different hydrogen concentrations: 0, 0.005, and 0.01 atomic ratios. Blue indicates an MSD value of 0.03 Å^2^, and red indicates an MSD value of 0.07 Å^2^.

**Figure 8 materials-13-04949-f008:**
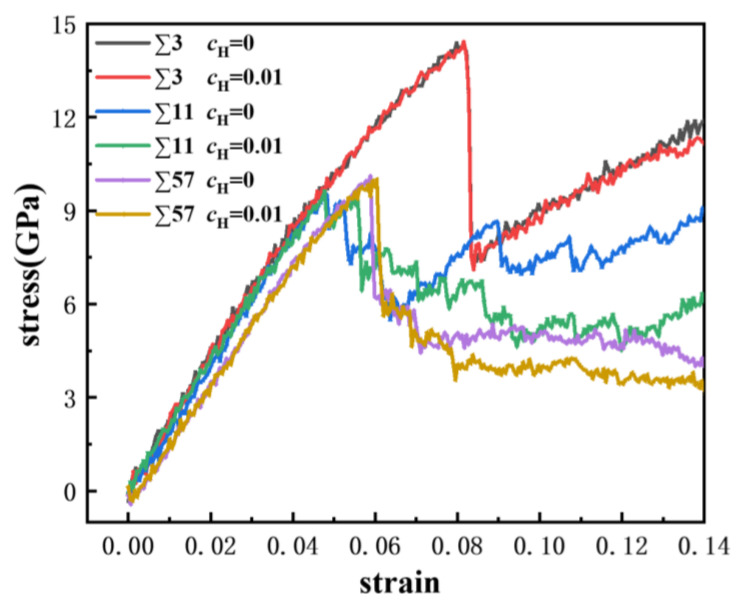
Comparison of strain-stress curves in different GBs at atomic hydrogen concentrations of 0 and 0.01.

**Figure 9 materials-13-04949-f009:**
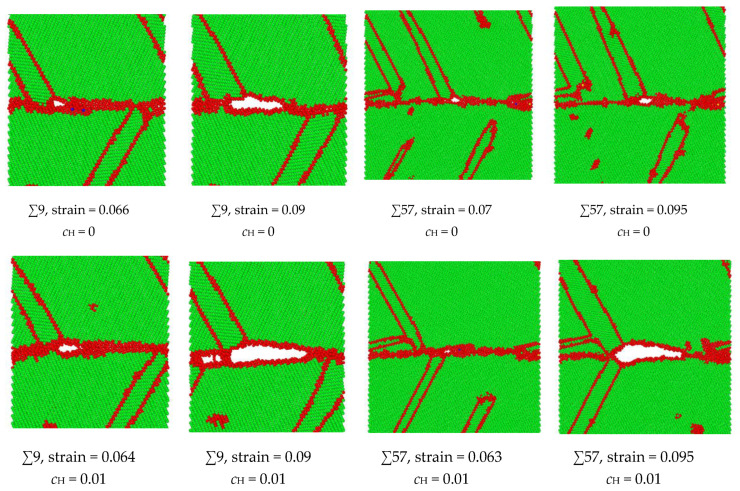
Comparison of intergranular crack geometries with or without H in ∑9 and ∑57 GBs. The green indicates BCC structure, and the red indicates unknown structures, e.g., grain bouddaries, free surfaces, or twinnings.

**Figure 10 materials-13-04949-f010:**
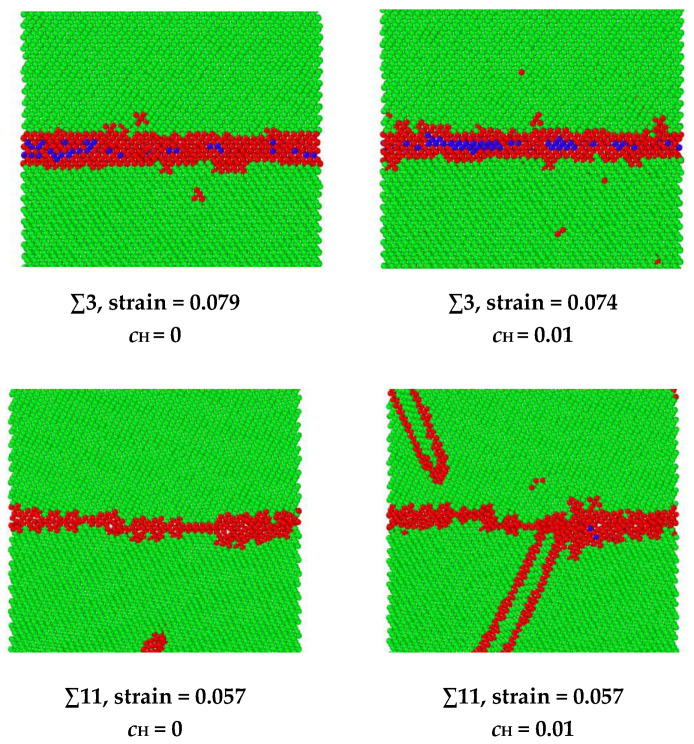
A comparison of the phase transition strain in ∑3 GB and twinning emission strain in ∑11 GB. The FCC structure is indicated in blue, the BCC structure is indicated in green, and the red indicates unknown structures.

**Figure 11 materials-13-04949-f011:**
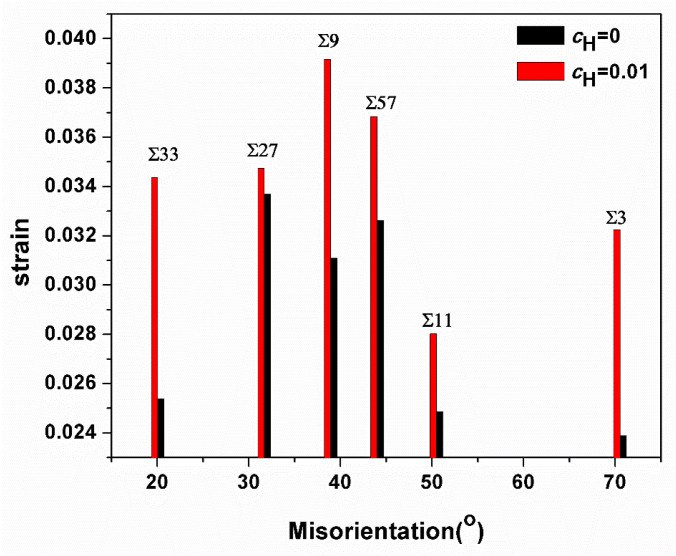
Comparison of the strain values at which first dislocations are emitted from the GBs.

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
