# Peer review of "Molecular Dynamics Studies of Hydrogen Effect on Intergranular Fracture in α-Iron"

_materials, 2020, doi:10.3390/ma13214949_

Round 1

Reviewer 1 Report

I find the paper interesting to read. Finding and explanations are well presented and therefore I suggest publication provided rectifying the following issues:

1) Boundary condition during load is stress free in x and z as explained in line 88. How about shear stress components? What magnitude they reach during the loading?

2) Information about the size of the system, number of atoms, program used for simulations, program used for visualization, thermostat and barostat damping factors are missing

3) Line 92, what is the result of the fit for H and Fe interactions? How good are they?

4) Line 104, equation 1 and 2, size dependency is studied?

5) why only three data points are included in Figure 5? More data points from other grains boundaries are required to make the argument in line 157 stronger.

6) Is there any particular reason only results of three grain boundaries are shown in Figure 8 while in Figure 6 and 7 there are more?

7) Minor issues:
line 121, "is shown" instead of "had been shown"
line 288 $1.9*10^{-30}$ should be 1.9x10^{-30}. Similar holds for all the numbers in the paper, e.g line 313 to 320.

Author Response

Dear Editor and Reviewers,

Thank you very much for the letter and the referees’ reports on our manuscript (materials-971405) titled “Molecular Dynamics Studies of Hydrogen Effect on Intergranular Fracture in α-Iron”. We are pleased to see some rather positive feedback on this manuscript. In response to these constructive suggestions and comments, all changes in the revised manuscript are highlighted in yellow. Our responses and the list of changes are also itemized below, following the original comments.

Reviewer #1:

Reviewer Comment: I find the paper interesting to read. Finding and explanations are well presented and therefore I suggest publication provided rectifying the following issues:

Thank you very much for your positive comment.

The reviewer wrote: “1. Boundary condition during load is stress free in x and z as explained in line 88. How about shear stress components? What magnitude they reach during the loading?

When the grain boundary was generated, the model is shear stress free. When the bi-crystal model is loaded, the loading style is Mode I. Hence, the shear stress should be neglected during loading.   

The reviewer wrote: “2. Information about the size of the system, number of atoms, program used for simulations, program used for visualization, thermostat and barostat damping factors are missing.”

Thanks for the comment. “Simulation Cells that were comprised of 54,000 to 58,000 Fe atoms were (97~107)×(193~227)×28 Å3 in three dimensions. All the simulations were carried out with LAMMPS, and OVITO was applied for visualization.” MD simulations were conducted using isothermal-isobaric (NPT) ensemble via Nose-Hoover thermostat. The manuscript has been revised accordingly.

The reviewer wrote: “3. Line 92, what is the result of the fit for H and Fe interactions? How good are they?"

“The fitted Fe-H potential had been applied to calculate the free surface energy, stacking fault energy, and strain-stress curves. The simulation results matched well with experimental tests. Moreover, the fitted potential had been verified by several previous works.” This description is inserted into the manuscript.

The reviewer wrote: “4. Line 104, equation 1 and 2, size dependency is studied?"

The ∑3 GB possessed the lowest volume expansion and GBE value simultaneously, which suggests that it is the most stable GB when no hydrogen is present. The ∑57 GB possessed the largest volume expansion and GBE. However, the comparison also suggests that grain boundary expansion is not necessarily positively related to grain boundary energy, and no linear correlation has been observed in Fig. 2. As hydrogen atoms were inserted into the GBs, the size expansion had been verified in all GBs.

Figure 2. Correlation between GBE and volume expansion for different symmetric GBs.

The reviewer wrote: “5. why only three data points are included in Figure 5? More data points from other grains boundaries are required to make the argument in line 157 stronger."

Thanks for the suggestion.  More data points from other GBs have been added to verify the conclusion that the correlation verifies that H diffusivity in GB is positively related to the extra volume, and hydrogen atoms trapped in a smaller excess volume are unstable and have higher potential energy.

  • (b)

Figure 5. (a) the correlation of the extra volume of specific GB versus the average of potential energy of H atom in it; (b) the correlation of the extra volume of specific GB versus the diffusivity of H atom in it.

The reviewer wrote: “6. Is there any particular reason only results of three grain boundaries are shown in Figure 8 while in Figure 6 and 7 there are more?"

The strain-stress curves of different GBs have been shown below. Because there are too many curves, overlapping occur in the figure. To depict more details of strain-stress curves, three representatives GBs were selected in the manuscript. The ∑3 GB possesses the smallest excess volume, the ∑11 GB possesses the medium excess volume, and ∑57 GB possesses the largest excess volume.

CL Figure 1. The strain-stress curves for different GBs.

The reviewer wrote: “7. Minor issues: line 121, "is shown" instead of "had been shown" line 288 $1.9*10^{-30}$ should be 1.9x10^{-30}. Similar holds for all the numbers in the paper, e.g line 313 to 320."

Thanks for the comments. The manuscript has been revised accordingly.

Sincerely,

Dr. Zili Li

College of Pipeline and Civil Engineering

China University of Petroleum (East China)

[email protected]

Reviewer 2 Report

This is a very thorough and inspiring study. I have a suggestion for the authors: extending the explanation of the meaning of the ∑ variable would greatly benefit the manuscript. This is a key parameter which is used in several places of the text/figures. As it is at present, the manuscript is not very accessible from non-specialized researchers who, nonetheless, could find some of the findings useful for their own purposes.

A few minor comments:

Line 112: Fig1 -> Fig.2

Fig.6 appears truncated on the right-hand side.

Line 194 “Since the color of GB changes from most blue (0.02 Å3) to partially yellow or red (0.07 Å3) after hydrogen is inserted.” I believe that this sentence is incomplete.

Author Response

Dear Editor and Reviewers,

Thank you very much for the letter and the referees’ reports on our manuscript (materials-971405) titled “Molecular Dynamics Studies of Hydrogen Effect on Intergranular Fracture in α-Iron”. We are pleased to see some rather positive feedback on this manuscript. In response to these constructive suggestions and comments, all changes in the revised manuscript are highlighted in yellow. Our responses and the list of changes are also itemized below, following the original comments.

Reviewer #2:

Reviewer Comment: This is a very thorough and inspiring study. I have a suggestion for the authors: extending the explanation of the meaning of the ∑ variable would greatly benefit the manuscript. This is a key parameter which is used in several places of the text/figures. As it is at present, the manuscript is not very accessible from non-specialized researchers who, nonetheless, could find some of the findings useful for their own purposes.

Thank you very much for your positive comment.

The reviewer wrote: “Line 112: Fig1 -> Fig.2

Fig.6 appears truncated on the right-hand side.

Line 194 “Since the color of GB changes from most blue (0.02 Å3) to partially yellow or red (0.07 Å3) after hydrogen is inserted.” I believe that this sentence is incomplete."

Thanks for the comments. The manuscript has been revised accordingly, and the revised section has been highlighted in yellow.

Sincerely,

Dr. Zili Li

College of Pipeline and Civil Engineering

China University of Petroleum (East China)

[email protected]

Reviewer 3 Report

The article has a high scientific interest. Hydrogen is one of the elements that has the most influence on the intergranular fracture of alpha iron. This study provides a very interesting mechanistic framework for intergranular failure. It is very well planned, and the results obtained are very interesting.
Regarding the theoretical model that it develops, to predict the intergranular cracking rate, we can say that it is a simple but very practical model. It is easy to use and offers good results.
It is clear that hydrogen induces intergranular cracking, so its study is essential to predict the behavior of this material.

It is of sufficient quality to be published. The conclusions should be reviewed and completed.

Author Response

Dear Editor and Reviewers,

Thank you very much for the letter and the referees’ reports on our manuscript (materials-971405) titled “Molecular Dynamics Studies of Hydrogen Effect on Intergranular Fracture in α-Iron”. We are pleased to see some rather positive feedback on this manuscript. In response to these constructive suggestions and comments, all changes in the revised manuscript are highlighted in yellow. Our responses and the list of changes are also itemized below, following the original comments.

Reviewer #3:

Reviewer Comment: The article has a high scientific interest. Hydrogen is one of the elements that has the most influence on the intergranular fracture of alpha iron. This study provides a very interesting mechanistic framework for intergranular failure. It is very well planned, and the results obtained are very interesting.

Regarding the theoretical model that it develops, to predict the intergranular cracking rate, we can say that it is a simple but very practical model. It is easy to use and offers good results.

It is clear that hydrogen induces intergranular cracking, so its study is essential to predict the behavior of this material.

It is of sufficient quality to be published. The conclusions should be reviewed and completed.

Thank you very much for your positive comment. The grammar and tempo of this article have been further improved.

Sincerely,

Dr. Zili Li

College of Pipeline and Civil Engineering

China University of Petroleum (East China)

[email protected]

Round 2

Reviewer 1 Report

I am satisfied with the authors modifications and response.